# Turning the Tables: Biased, Imbalanced, Dynamic Tabular Datasets for ML Evaluation

**Sérgio Jesus**[1,2], **José Pombal**[1], **Duarte Alves**[1], **André F. Cruz**[1],
**Pedro Saleiro**[1], **Rita P. Ribeiro**[2,3], **João Gama**[3], **Pedro Bizarro**[1]

[1]Feedzai
[2]DCC Faculdade de Ciências da Universidade do Porto
[3]LIAAD, INESCTEC Universidade do Porto
`{sergio.jesus, pedro.saleiro}@feedzai.com`

## Abstract

Evaluating new techniques on realistic datasets plays a crucial role in the development of ML research and its broader adoption by practitioners. In recent years, there has been a significant increase of publicly available unstructured data resources for computer vision and NLP tasks. However, tabular data — which is prevalent in many high-stakes domains — has been lagging behind. To bridge this gap, we present **B**ank **A**ccount **F**raud (**BAF**), the first publicly available[1] privacy-preserving, large-scale, realistic suite of tabular datasets. The suite was generated by applying state-of-the-art tabular data generation techniques on an anonymized, real-world bank account opening fraud detection dataset. This setting carries a set of challenges that are commonplace in real-world applications, including temporal dynamics and significant class imbalance. Additionally, to allow practitioners to *stress test* both performance and fairness of ML methods, each dataset variant of BAF contains specific types of data bias. With this resource, we aim to provide the research community with a more realistic, complete, and robust test bed to evaluate novel and existing methods.

## 1 Introduction

The ability to collect and handle large-scale data has laid the foundations for the widespread adoption of Machine Learning (ML) [1, 2]. Regardless of the application, evaluating new ML techniques on realistic datasets plays a crucial role in the development of ML research, and subsequent adoption by practitioners [3] [4]. Additionally, with the growing ethical concerns around the potential of bias in algorithmic decision-making [5–7], fairness evaluation is becoming a standard practice in ML [8–10]. However, the vast majority of publicly available datasets are directed to computer vision and NLP tasks, and there is a scarcity of large-scale domain-specific tabular datasets. The latter are the centerpiece of most high-stakes decision-making applications, where fairness testing is of paramount importance. As it stands, the most relevant tabular datasets in the Fair ML literature suffer from a series of limitations [11–13], which we will detail in Section 2. Furthermore, most real-world settings are dynamic, featuring temporal distribution shifts, class imbalance, and other phenomena that are not reflected in most of the datasets in Fair ML literature [14]. We will discuss how the Bank Account Fraud (BAF) suite of datasets tackles these limitations, and outline its utility as a general-purpose tool for the evaluation of performance and fairness in dynamic environments.

---

[1]Available at `https://github.com/feedzai/bank-account-fraud`

36th Conference on Neural Information Processing Systems (NeurIPS 2022) Datasets and Benchmarks track.

**What is a good dataset for ML practitioners?**

In general, good datasets for ML benchmarks are ones that are representative of the distribution and dynamics of some target population, and that, symbiotically, are useful to train ML models for a given task. Large-scale datasets based on real-world use cases fulfill both goals, as they contain a wide variety of observations, and because findings from benchmarks conducted on them are considered to be sufficiently generalizable to real-world production environments [15].

Adding to these characteristics, a key aspect of a dataset for Fair ML is the context of the task: high-impact domains — where decisions produced by an ML system have substantial consequences on the lives of the decision subjects — are strongly preferred [9, 10]. Applications of this nature may be found in the criminal justice, hiring, and financial services domains, among others. Another important aspect for the community is the fidelity of the settings. That is, datasets originating from real-world scenarios are favoured, especially if ML methods are employed in such setting. In these cases, the impact of a new method can be measured and compared to other alternatives, or even to the original decision-making policy. These measurements can then be translated into real-world solutions, namely making models fairer with respect to a historically discriminated group. Other important components for these datasets include the available protected attributes, privacy, representation, scale, and recency.

**What is the current landscape of datasets for Fair ML research?**

Only a limited amount of datasets are consistently used for validating and benchmarking fairness methods. In fact, there is a trend of *funneling* in the ML community in general [16], with fewer datasets being used more often for experimental observations. Common issues regarding these datasets are expanded in section 2.1. The relative age of the majority of the datasets used in Fair ML, combined with the saturation of tests performed on them, makes the observed results stagnate. These constitute technical considerations for deprecating the dataset [17], and limit the possibility of validating novel solutions. The lack of quality datasets for Fair ML — identified in the 2021 Stanford University's AI Index Report [18] — has prompted the appearance of several initiatives advocating the public sharing of private datasets for decision-making containing protected attributes. Symbiotically, many tools have been recently developed which facilitate the sharing of data, namely on best practices in documentation and privacy-preserving methods [19, 20]. However, there is still no observable shift from the older datasets to the more recent, comparatively less explored datasets.

**What are the characteristics of the BAF suite?**

The BAF suite comprises six datasets that were generated from a real-world online bank account opening fraud detection dataset. This is a relevant application for Fair ML, as model predictions result in either granting or denying financial services to individuals, which can exacerbate existing social inequities. For instance, consistently denying individuals from one group access to credit may perpetuate or even widen existing wealth inequality gaps. Each dataset variant in the suite features predetermined and controlled types of data bias over multiple time-steps, which are detailed in Section 3.3. The aforementioned variants, combined with the temporal distribution shifts inherent to the underlying data distribution, amount to an innovative medium for *stress testing* the performance and fairness of ML models meant to operate in dynamic environments.

The datasets on the suite were generated by leveraging state-of-the-art Generative Adversarial Network (GAN) models [21]. One important reason for choosing these methods was to take into account the privacy of the applicants — an ever-growing concern in today's societal and legislative landscape [22]. Each dataset is comprised of a total of one million instances of individual applications, using a total of thirty features. The latter represent observed properties of the applications, either obtained directly from the applicant (*e.g.*, employment status), or derived from the provided information (*e.g.*, whether the provided phone number is valid), and aggregations of the data (*e.g.*, frequency of applications on a given zip code). The data spans eight months of applications, which can be identified in the column ''month''. Regarding protected attributes, the dataset provides the age, personal income, and employment status of the applicant. To provide some degree of differential privacy [23], we injected noise into the instances of the original dataset, and categorized personal information columns, such as income and age, prior to the training of the GAN model. More details on the dataset are included in Sections 3 and 4, and in the dataset's datasheet [19], provided as supplementary material.

We must also discuss possible shortcomings of the presented suite of datasets. One well-known issue present in most financial services domains is that of selective labelling [24], *i.e.*, we can only know

the true labels of the applicants that were accepted. As such, if an applicant is pre-screened and rejected by some rule-based or ML-based model, then we will never know its true label (*e.g.*, whether the applicant was actually fraudulent). Importantly, all labels present in the dataset correspond only to ground-truth data (pre-screened applicants are left out). In our case, the pre-screening process was minimal, consisting only of regulation checks (*e.g.*, anti-money laundering regulations may disallow banks from accepting certain customers), and business-oriented checks specifically for the credit card that was being offered (only 18+ years of old residents in a specific country in which the bank operates). Hence, although some selection bias is strictly inevitable due to banking regulations, the dataset comprises the widest pool of applicants possible.

## 2 Background

### 2.1 Shortcomings of Popular Tabular Datasets

To the best of our knowledge, there are no public large-scale bank account opening fraud datasets. That said, there are two relevant datasets in the more general banking fraud domain, both pertaining to transaction fraud. The first [25] is based on data from about 300k transactions from European credit card holders for the period of September 2013. Its fundamental drawback is that the features of the dataset are principal components of the original features (with generic names like V1, V2, etc. . . ), leaving no useful real-world information for the users, and making it impossible to study algorithmic fairness. The second dataset [26] is in fact a suite of 5 synthetic datasets, based on a real-world mobile transaction fraud use case; they are reasonably large, and contain informative feature names. However, the data contains no sensitive attributes, limiting its use in the context of algorithmic fairness. In any case, our suite of datasets would still be a valuable contribution, since it is based on a different fraud application. Bank account opening makes for a particularly important use-case to benchmark Fair ML methods, as opening a bank account is essential in today's society, and gate-keeping such a service can seriously hinder the well-being of an individual [27].

Among the datasets used for the benchmark of fairness methods, the UCI Adult dataset [28, 29] stands out as the most popular in the field [11, 30]. Despite its popularity, the dataset has recently been criticized [11, 12], mainly due to three aspects: a) the sampling strategy, based on the poorly documented variable `fnlwgt`, b) the arbitrary choice of task — predicting individuals whose income is above 50,000 dollars — which is not connected to any real census task, and c) the age of the data itself (it is based on 1994 US census data).

Similar issues are found on a variety of datasets. For instance, the second most popular dataset for fairness benchmarks, COMPAS [6], a risk assessment instrument (RAI) dataset in the criminal justice domain. This dataset is afflicted with measurement biases [13], missing values, label leakage [11] and sampling incongruities [31]. Most importantly, in the domain of criminal justice there are usually more agents that can influence a given decision, apart from an ML model. Thus, measuring fairness in this domain solely based on ML model predictions often provides an unrealistic picture of the actual outcomes that take place in the real world. Additionally, one major concern is regarding the privacy of the data, as it is possible to identify accused individuals based on criminal record and other Personal Identifiable Information (PII) [11].

The third most popular dataset is the German Credit dataset [29], which has several documentation issues, including the information regarding what is used as sensitive attribute. Here, the sex of the individual is not retrievable by the ''`Personal status and sex`'' attribute, as there are overlaps between the possible values. A posterior release of the dataset addresses some documentation errors, but also confirms that retrieving the applicant's sex through the aforementioned attribute is not possible [32]. This limits the utility of the dataset in the context of algorithmic fairness. Additionally, the dataset is composed of applicants from 1973 to 1975, which hinders the generalization of any insights to today's world.

Recently, a study on the datasets used in Machine Learning Research (MLR) identified a funneling tendency in the field, whereby increasingly fewer datasets are being used for benchmarking [16, 11]. These datasets are generally also being used in different tasks than originally intended [16]. Such a trend is also observed in the fairness community, where the previously mentioned UCI Adult dataset was repurposed from its original task [28, 29]. This highlights the necessity of renewing the currently available datasets for Fair ML. Besides the aforementioned datasets, there is a study [33] on the characteristics of an additional 12 datasets used, albeit less frequently, in the Fair ML literature. We

point to this work for a more comprehensive analysis of each dataset, while maintaining that none of them fulfill all the desiderata listed in Section 1 (*e.g.*, they have small, unrealistic samples, are too old, are not based on a specific task, etc...).

## 2.2 Privacy-Preserving Approaches and Generative Models

A major concern regarding the publication of datasets is the rise of potentially dangerous privacy-breaching applications for the data [34]. To avoid this issue, it is required to either remove, transform, or obfuscate any information that leads to the identification of a particular individual.

One of the more consensual means of evaluation of methods for the purpose of privacy-preservation is the measurement of differential privacy [23, 35]. This metric determines the maximum difference in an arbitrary measurement or transformation applied to a dataset induced by any individual instance. Lower values of this metric correspond to higher preservation of privacy. Upper-bound levels of this metric are met on several generative models [36, 37]. However, the default implementations of generative models do not take into consideration common problems faced in the tabular data domain. These are mostly caused by having categorical and non-normally distributed continuous variables.

One particular architecture that tackles these problems is the CTGAN [38]. This architecture, however, does not have differential privacy guarantees by default, whereas models adapted to the image domain, for example, do; this constitutes a gap in generative models for tabular data. Recently, extensions to the CTGAN that are trained with differential privacy guarantees have been proposed (DPCTGAN and PATECTGAN [39]), with mixed empirical results in terms of the generated data's utility [40]. Another approach to promote privacy in generated tabular data is to add a noise mechanism to the original data prior to GAN training [41]. This way, the GAN never has access to specific applicant data. We take this approach for our suite of datasets, since the former methods would not output sufficiently informative data to be useful in practice.

There is still no consensus on the evaluation of generative models [42]. In the computer vision domain, most approaches present a measurement of distance between the original and generated data distributions, such as the Inception Score (IS) and Fréchet Inception Distance (FID) [42]. For tabular data, the practice revolves mostly around validating the generated data through training models on the combination of the generated and original datasets [38, 43], analyzing statistics derived from distance between individual feature distributions, and computing paired correlations [43].

# 3 The BAF Dataset Suite

## 3.1 Original Dataset Overview

The introduced datasets regard the detection of fraudulent online bank account opening applications in a large consumer bank. In this scenario, fraudsters will attempt to either impersonate someone via identity theft, or create a fictional individual in order to gain access to the banking services. After being granted access to a new bank account, the fraudster quickly maxes out the accompanying line of credit or uses the account to receive illicit payments. All costs are sustained by the bank, as there's no way of tracing the fraudster true identity.

Our use case is considered a high-stakes domain for the application of ML. A positive prediction (*i.e.*, flagged as fraudulent) leads to a rejection of the customer's bank account application (a punitive action), while a negative prediction leads to granting access to a new bank account and its credit card (an assistive action). As mentioned in Section 1, holding a bank account is a basic right in the European Union [27], making fraud detection an extremely pertinent application from a societal perspective. Following the recent awareness of the risk of unfair decision-making using ML systems, banks and merchants are in a front-line position to become early adopters of Fair ML methods. Nonetheless, a potential drop of a few percentage points in predictive performance often represents millions in fraud losses, making the requirements for Fair ML particularly stringent.

Each instance (row) of the dataset represents an individual application. All of the applications were made in an online platform, where explicit consent to store and process the gathered data was granted by the applicant. The label of each instance is stored in the `"is_fraud"` column. A positive instance represents a fraudulent attempt, while a negative instance represents a legitimate application. The dataset comprises eight months of information ranging from February to September. The prevalence

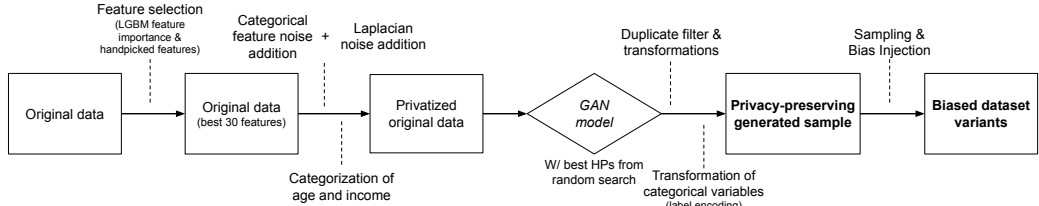

Figure 1: Overview diagram of the steps to obtain the datasets of the suite.

of fraud varies between $0.85\%$ and $1.5\%$ of the instances over different months. We observe that these values are higher for the later months. Additionally, the distribution of applications also changes from month to month, ranging from $9.5\%$ on the lower end to $15\%$ on the higher end. These distributions are reference in order to define the approximate number of legitimate and fraudulent instances that should be sampled each month for each variant of the dataset in the suite.

## 3.2    Training and Validating a Generative Model with Privacy Concerns

The first step of this process was to reduce the number of original features in the dataset. This has two main consequences; firstly, we improve the convergence time and results of the generative model. Secondly, we mitigate some privacy issues on the resulting dataset, since there is less available information for tracing applicants (although privacy is still a concern). To this end, we started by selecting the five best performing LightGBM [44] models in the original dataset, out of 200 hyperparameter configurations, obtained through random search over a parameter grid. Then, we selected the union of the top thirty most important features for these five models, according to the default feature importance method of LightGBM (number of splits per feature in the model). This resulted in a total of forty three features. This selection was reduced further to thirty features, by selecting more expressive, interpretable, and less redundant features manually. Additionally, we perform a study on the overlap between the selected features from the LightGBM models and other commonly used models, in particular by running a similar study with Random Forests, Decision Trees and Logistic Regressions. Results of this study are available in Appendix (Section A.7).

To enforce a differential privacy budget $\epsilon$ in the datasets, we perturb each column in the original dataset using a Laplacian noise mechanism [41], prior to training the GAN. According to this formulation, the noise level is inversely proportional to the privacy budget. The results of this analysis are available in Appendix A.4 — we selected a privacy budget that obtained a reasonable trade-off between data privacy and utility. To further obfuscate the dataset, the information regarding applicant's age and income was categorized based on the value and quantile, respectively. Thus, the GAN model never has access to specific applicant data.

Afterwards, we trained the CTGAN models on the perturbed dataset with the selected features. Since there are no generative model architectures capable of modeling temporal data out-of-the-box, we add this functionality by creating a column representing the month where the application was made. We found this segmentation to be a good trade-off between sample size and granularity. The selection of hyperparameters for the generative model was done through random search, resulting in a total of 70 trained CTGAN models. The tested hyperparameters are available in Appendix (Section A.1). Generative models were trained in parallel, in four Nvidia GeForce RTX 2080 Ti models. The average (non-parallelized) time to train a single generative model was of 4.53 hours, totaling in close to 13 days of computation time. For each instance, we encoded a single unique identifier depending on the feature values, so that there could be no repetitions between the original data and the generated datasets, or among the generated datasets, further promoting privacy and diversity within the suite.

With the aforementioned setup, we created samples from the generative model of 2.5M instances. We then reduced these samples to candidate datasets with 1M instances by further sampling observations, such that the observed month distribution and prevalence by month corresponded approximately to those of the original dataset. These datasets would then be evaluated to assess the quality of the generated model (for more details on the results see the Appendix, Section A.3). The first group of metrics pertains to the predictive performance of ML models on combinations of data. This extends previous works [38, 43]. In these, the trained models use generated data in train and are tested on real

data. We extend this methodology by also training with real data and testing on generated data, and training and testing using exclusively generated data (we maintain a step on training on generated data and testing on real data). The second group of metrics is on the statistical similarity between the real and generated data. We calculate the average absolute difference in Pearson correlation between the real data and the generated data [43] over all features, as well as the average distance between the empirical cumulative distribution functions of each feature for the datasets. We additionally present the Spearman correlation between every feature and the target variable. These measurements are present in Sections A.3 to A.7 in the Appendix. The goals behind leveraging both sets of metrics are to make sure that models trained on the generated datasets are effective at the task at hand, and to guarantee that the generated distribution is realistic and faithful to the original data.

Figure 1 contains an overview diagram of the generative process for the datasets in the suite.

During the process of training a generative model, as well as obtaining the empirical observations, several choices were made. These are listed and justified bellow.

**Splitting Strategy:** Leveraging the ``month'' column, we were able to split the data temporally: the first six months for training, the last two for testing. This is common practice in the fraud domain — and the strategy used with the original dataset — as more recent data tends to be more faithful to the data's distribution when models are put in production.

**Protected Attributes:** The dataset includes three relevant features that are possible to use as protected attributes for the data: "customer_age", "income" and "employment_status". In this study, we focus on customer age, for which we present the original and generated distributions in Appendix (Section A.6). To be able to compute group fairness metrics, we create a categorical version by separating applicants with age $>50$ in one group and $\leq 50$ in the other group.

**Performance Metric:** Due to the low prevalence figures in the data, it is important to define a relevant threshold and metric for the application. This is done mainly through defining a specific operating point in the ROC space of the model. In this case, we select the threshold in order to obtain 5% false positive rate (FPR), and measure the true positive rate (TPR) at that point. This metric is typically imposed by clients in the fraud detection domain, since it strikes a balance between detecting fraud (TPR), and keeping customer attrition low — each false positive is a dissatisfied customer that may wish to change the banking company after being falsely flagged as fraudulent.

**Fairness Metric:** In this scenario, a penalizing effect for an individual would be a wrongful classification for a legitimate applicant, *i.e.*, a false positive. Because of this, for the context of fairness, we want to guarantee that the probability of being wrongly classified as a fraudulent application is independent of the sensitive attribute value of the individual. Hence we measure the ratio between FPRs, *i.e.*, *predictive equality* [45]. The ratio is calculated by dividing the FPR of the group with lowest observed FPR with the FPR of the group with the highest FPR.

### 3.3 Bias Patterns

To further enhance its generalization capabilities, namely to *stress* test predictive performance and fairness, the suite contains six datasets (variants of the base dataset) each one with pre-determined and controllable bias patterns. The data biases we introduced are based on a data bias taxonomy proposed in previous work [46], as follows:

**Group size disparity** is present if $P\left[A = a\right] \neq \frac{1}{N}$, where $a \in A$ represents a single group from a given protected attribute $A$, and $N$ the number of possible groups. This represents different group-wise frequencies in the dataset, and might be caused by numerous reasons, such as an original population with imbalanced groups, or uneven adoption of an application by demographic segments. Considering the example of the presented dataset, where age is the protected attributed, group size disparity would imply that age groups have different sizes. This pattern is observed in the original dataset, with a higher proportion of applications being made by the younger age group.

**Prevalence disparity** occurs when $P\left[Y\right] \neq P\left[Y|A = a\right]$, *i.e.*, the class probability depends on the protected group. We leverage this property to generate datasets whose probability of the label is conditioned by the different groups of the protected attribute. Similarly to the original dataset, the proposed dataset shows higher fraud rates for older age groups. The reason for this might be because fraudsters have an incentive to impersonate older people: banks provide older applicants with larger lines of credit once an account is opened, which fraudsters try to max out before being caught.

**Separability disparity** extends the previous definition by including the joint distribution of input features $X$ and label $Y$, $P[X, Y] \neq P[X, Y|A = a]$. An example of this, consider an ATM withdrawal scenario, where we have a binary feature (illumination) indicating if the ATM has external light close by, and age. Also, suppose that the age group 20-40 has a higher probability of using ATMs in dark places. This leads to a greater likelihood of having their card cloned by a fraudster. The illumination feature will help identify fraud instances for records within that group, but not for the remaining instances.

The first and second disparities are induced through controlling the generative model sampling, depending on the group and label, respectively. Inspired by previous approaches [47], the third disparity is obtained through appending two columns with different multivariate normal distributions, whose means depend on the group and label, with different controllable linear separability.

## 3.4 BAF Variants

The BAF suites contains one base dataset and 5 additional dataset variants with additional controlled data bias patterns. Each dataset variant follows the same underlying distribution as the base dataset, i.e., each instance of the suite is sampled from the same generative model. This implies that, save for prevalence and group disparities in some cases, whatever biases were present in the base dataset are also present in the variants. The goal is to offer a diverse set of additional algorithmic fairness challenges. A summary of the generated variants can be found in Table 1.

Table 1: Summary table of the generated variants in the study. Approximate values for the original dataset. Values in parentheses are applied to the test set.

| Dataset | Group | Group Size | Prevalence | Separability |
|---------|-------|-----------|-----------|--------------|
| Original | Majority | 80% | 1% | - |
|          | Minority | 20% | 2% | - |
| Base | Majority | 77% | 0.9% | - |
|      | Minority | 23% | 1.8% | - |
| Variant I | Majority | 90% | 1.1% | - |
|           | Minority | 10% | 1.1% | - |
| Variant II | Majority | 50% | 0.4% | - |
|            | Minority | 50% | 1.9% | - |
| Variant III | Majority | 50% | 1.1% | Increased |
|             | Minority | 50% | 1.1% | Equal |
| Variant IV | Majority | 50% | 0.3% (1.5%) | - |
|            | Minority | 50% | 1.7% (1.5%) | - |
| Variant V | Majority | 50% | 1.1% | Increased (Equal) |
|           | Minority | 50% | 1.1% | Equal (Equal) |
| Global | - | - | 1.8% | - |

**Variant I.** Contrary to the Base and Original datasets, the groups in the protected attribute of this variant do not have disparate fraud rates. Instead, the group size disparity is aggravated, reducing the size of the minority group from approximately 20% of the dataset to 10%. As such, while models trained on this dataset will not face the challenge of group-wise prevalence imbalance, they still have to be robust to the fact that there is an even smaller minority group, which may be left under-explored.

**Variant II.** This variant features steeper prevalence disparities than Variant I and base — one group has five times the fraud rate of the other, instead of approximately two times — while group sizes are equal. Thus, this variant serves as a *stress test* for the prevalence disparity bias.

**Variant III.** This dataset features the Separability disparity presented in Section 3.3, whereby the classification task is made relatively simpler for the majority group by manipulating the correlations between the protected attribute, appended features, and the target. This type of bias calls for more nuanced interventions; for instance, re-sampling the data to balance prevalence and group size is ineffective, as they are already balanced. Thus, for models to be fair and stay performant under this

variant, it is important to reach an equilibrium between countering the relations among some features and the protected attribute, while still learning useful patterns.

**Variant IV.** This variant introduces a temporal aspect to the presented data biases. In particular, similar to Variant II, it features prevalence disparities over the first six months, but no disparity for the remainder. Considering the first six months as a training set, and the rest as validation data, the observed disparity can be caused by a biased training data collection process, for example. Taking such aspects into account is fundamental to model realistic dataset variants, since real-world use cases are susceptible to biases outside of the practitioner's immediate control, and that change across time.

**Variant V.** Similar to the previous variant, this dataset features changes in data bias patterns over time. However, we keep group-size and prevalence balanced. Instead, we add a separability bias component on the first six months, and remove it on the remainder. This is essentially a feature distribution shift across time, where we make sure that the features that change are related to both the protected attribute and the target. Most models in the real-world operate in highly dynamic environments, which makes them highly susceptible to temporal distribution shifts. In fact, this variant is analogous to a very common phenomenon in fraud detection: fraudsters adapting to the outcomes. That is, fraud detection is an adversarial classification setting [48] (a subset of performative prediction [49]), where fraudsters may adapt their behaviour over time to evade detection. This means that features that were useful to detect fraud for a time, may become obsolete afterwards, as fraudsters learn to escape the system. In Variant V, these features are related to the protected attribute and the target, which can lead to drastic changes in the landscape of performance-fairness trade-offs [50].

## 4 Empirical Observations

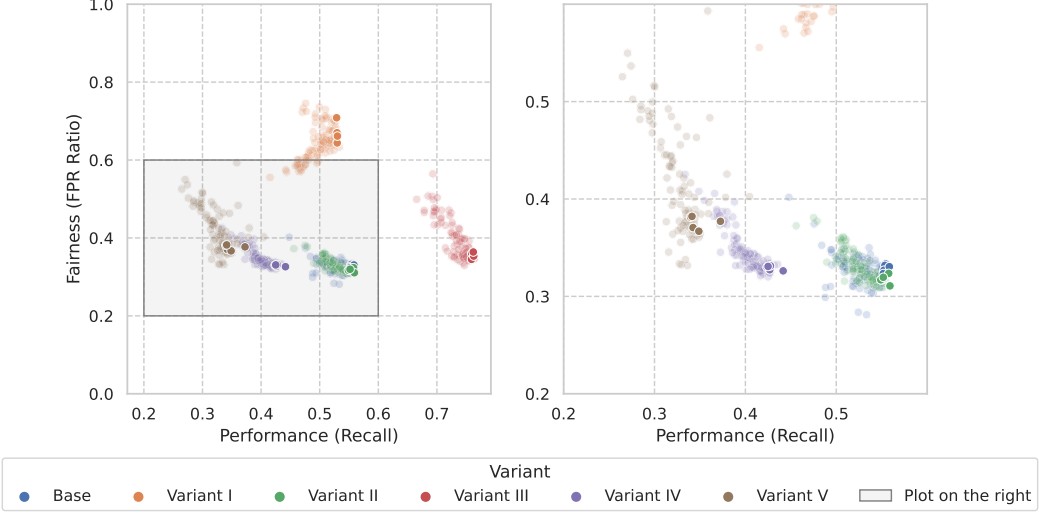

Figure 2: On the left, fairness and performance of 100 LightGBM models across all datasets in the suite. On the right, a zoom-in that focuses on the base dataset and Variant II, compared with the variants that feature temporal bias (IV and V). Opaque points represent the top 5 models in terms of performance (TPR) in the Base dataset, across all variants. The top performing models on the Base dataset are not necessarily the best ones on the other variants.

To paint a teaser picture of the performance and fairness challenges that practitioners would face using our suite, we assessed how fairness-blind models fared on each dataset. To this end, we sampled 100 hyperparameter configurations of LightGBM — a popular algorithm for tabular data — and trained them on each dataset. We measure performance as TPR at 5% FPR, as explained in Section 3. Our fairness metric is *predictive equality* (ratio of group FPRs), which makes sure that legitimate individuals of particular groups are not disproportionately denied access to banking services. This metric is appropriate for our *punitive* setting [45], as a positive classification translates into denial of banking services. That said, we strongly encourage practitioners to explore other fairness and performance metrics, as well as fairness-aware models on these datasets.

Figure 2 shows the fairness and predictive performance of all the models evaluated on the test set, using the first 6 months for training and the rest for testing. One pattern that stands out is that models are distributed in significantly different areas of the fairness-accuracy space, depending on the dataset they were used on. This is promising in terms of our goal of providing the community with a diverse suite of datasets. Additionally, the base dataset alone provides a demanding fairness challenge, with the top performing models lying around 0.3 FPR ratio. This implies that legitimate applications from individuals in the group of higher ages are more than three times more likely to be flagged as fraudulent, when compared to the group of lower ages.

Focusing on the variants, many models produced fairer results under Variant 1, when compared to the baseline. Still, there is more variance in the fairness axis, leaving room for improvement. Fairness of models under Variant II variant matches the baseline; this comes as a surprise, since the disparity in group fraud rates is larger in Variant II, which is expected to have consequences on fairness. With the appended features to induce the separability bias, models under Variant III were able to increase performance, at a comparable level of fairness of the base dataset and Variant II.

As for the variants with biases that change across time, there are some interesting findings. Looking at Figure 2, model performance deteriorated under the Variant IV variant, relative to its counterpart Variant II. The fact that the learned patterns in the training set do not carry over to the test set (like in Variant II) explains this gap in performance. The same reasoning applies to models under Variant V, which, compared to those under Variant III, show a similar, yet much more pronounced performance degradation phenomenon, and no gains in fairness. The plot on the right in Figure 2 shows how the best performing models under the baseline dataset were not necessarily the best ones, especially after introducing temporal biases (Variant IV and Variant V datasets). In fact, several other models achieved better fairness-accuracy trade-offs under these datasets. This shows how performant models in static environments may fall short in more realistic, dynamic ones.

All in all, the proposed suite seems to be an adequate tool to benchmark the fairness and performance of ML models meant for static and dynamic environments. We limited our analysis to fairness-blind models hoping that this encourages practitioners to experiment with other alternatives, including fairness-aware methods.

## 5   Limitations and Intended Uses

We identify two main challenges regarding the suite of datasets. The first regards guarantees of differential privacy. The fact that the original data is composed of aggregation features, and that the published suite is synthetically generated, the re-identification of individuals is not trivial. That said, it is still a potential issue if no theoretical privacy guarantees can be given. With this in mind, we applied Laplacian noise to the original data's features prior to training, which ensures a degree of privacy that depends on the privacy budget [41]. We chose a dataset that achieved an acceptable data privacy and utility trade-off. Furthermore, we categorical encoded continuous features that could reveal significant personal information if used directly (applicant income and age). Additionally, we made sure no generated instance matched exactly an original instance. Despite offering more guarantees, methods that include differential privacy constraints into the training process of the GAN — such as PATECTGAN and DPCTGAN [36, 51, 39] — did not yield good results in terms of data utility. This motivates us to explore ways to improve on this issue in future work.

The other challenge is related to the method of obtaining information. Many of the fields in applications were filled by the applicant. This might lead to wrongful information, either provided intentionally by fraudsters to boost their chances of success, or accidentally by legitimate applicants. To the best of our knowledge, there is no solution to this problem.

There are several possible uses for this suite of datasets. We note, however, that this dataset should only be used for the purpose of evaluating ML methods and Fair ML interventions, as the patterns and behaviours of banking fraud are highly dynamic and context-dependant. Models trained on this data should not be directly employed in real-world fraud detection scenarios, with the potential risk of under-performing or outputting biased decisions.

In this study, we limited our analysis to the original data split, *i.e.* training models with the initial 6 months of data, and testing on the remainder. These, however, can and should be adapted to other scenarios, which would confer more realistic and robust results *e.g.*, having part of the data for

validation of the hyperparameters or threshold definition, or having a sliding window approach to train and validate models. Additionally, we defined a threshold for the studied protected attribute (age), at the value of 50. We selected this value as it represents a decent compromise between group size (approximately an 80/20 split) and prevalence (approximately 2 times larger for the older group). This threshold, however, is not intended to be mandatory; other thresholds or group definitions should be taken into consideration.

We encourage other authors and practitioners to experiment with different ML or Fair ML algorithms on this suite of datasets. We expect that with this work, the quality of evaluation of novel ML methods increases, potentiating the development of the area. Additionally, we hope it encourages other similar relevant datasets to be published from other authors and institutions.

## Acknowledgments

We would like to thank Catarina Belém, João Bravo, João Veiga, Ricardo Moreira and Prof. Bruno Cabral for their invaluable contributions to this work.

The project CAMELOT (reference POCI-01-0247-FEDER-045915) leading to this work is co-financed by the ERDF - European Regional Development Fund through the Operational Program for Competitiveness and Internationalisation - COMPETE 2020, the North Portugal Regional Operational Program - NORTE 2020 and by the Portuguese Foundation for Science and Technology - FCT under the CMU Portugal international partnership.

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
