# OpenReview forum: "Turning the Tables: Biased, Imbalanced, Dynamic Tabular Datasets for ML Evaluation"
_NeurIPS.cc/2022/Track/Datasets_and_Benchmarks — NeurIPS 2022 Datasets and Benchmarks _

### Official Review · Reviewer_GZkf · 2022-07-19
**New dataset for evaluating ML methods and fair ML interventions**

**Rating:** 7
**Confidence:** 3
**Correctness:** The construction of the data set is s…
**Clarity:** The paper is very well written and ea…

**Strengths:**

The generated data sets address a gap wrt existing benchmarks. These were generated based on a real-world bank account opening application data set, and the properties of the generated data were preserved wrt to the original one.  The different variants generated, correspond to common problems in the real-world, like distribution shifts.  Hence it is expected that these data will be used for evaluating fairness of ML methods in a relevant and realistic scenario.

**Weaknesses:**

The limitations of the work are well outlined: (i) There are no theoretical guarantees on privacy as there is no upper bound limit for the metric of differential privacy, and (ii) the information was provided by applicants which could be, intentionally or not, wrong.

The authors also point out that the data set is only meant for evaluating fairness, but cannot be directly employed to real-world fraud detection scenarios.  Given the highly dynamic context here, this was to be expected.

**Additional Feedback:**

none

**Documentation:**

The documentation is sufficient.  However, it would be also important to not only state the months (February - September) during which the real-world data set was collected, but also the year.  The authors rightly point out that some of the existing data sets are too old.  It is important that one has also clarity wrt how recent the presented data set is.

**Ethics:**

There are no ethical concerns.

**Relation To Prior Work:**

The shortcomings of existing data sets has been discussed, which include:
- age of the data itself
- measurement bias
- missing values
- label leakage
- use of data sets for a purpose they were not intended for initially.


**Summary And Contributions:**

Various machine learning methods are used today for many high-stake decision making applications.  In such contexts algorithmic fairness is an important topic.  To test and evaluate ML methods wrt their fairness, benchmark data sets are needed.  While there exist numerous of these for computer vision and NLP tasks, there are only few for tabular data sets.  However, a number of high-stake decision making use cases, such as the considered one of detecting whether the online opening of a bank account was fraud or not, are based on tabular data.  This paper presents the first publicly available, large-scale, and privacy preserving suite of tabular data sets for fraud detection which is very valuable for evaluating ML methods and fair ML interventions.

---

> ### Author Response · Authors · 2022-08-25
> **We thank the author for the thorough review**
>
> Thank you for your thorough review. Regarding the raised issue:
>
> > However, it would be also important to not only state the months (February - September) during which the real-world data set was collected, but also the year. The authors rightly point out that some of the existing data sets are too old. It is important that one has also clarity wrt how recent the presented data set is.
>
> Due to client privacy reasons, we can not disclose the exact year of the dataset. We can confirm that the data is posterior to 2015; this constitutes a more recent dataset than most of the datasets used in Fair ML literature. You can observe a comprehensive list of datasets used for fairness in references [11] and [26]. The latter was added in the revised version of the paper.

---

### Official Review · Reviewer_Gpqi · 2022-07-23
**Synthetic Data for Bank Account Fraud**

**Rating:** 6
**Confidence:** 4

**Strengths:**

The paper is generally well-structured and well-written. It tackles an important problem and I could therefore see such benchmarks being useful for the broader community. I also like that the paper specifically addresses privacy concerns.

The paper motivates and proposes a fairness metric to go with it.I think this is important and relevant. I am however not entirely sure how fairness connects to fraud, I think this should be motivated more thoroughly, e.g. by providing an example scenario where fairness considerations are relevant.

I like that the paper proposes a metric and several concrete settings for study. This allows users to refer to specific studies. I am missing a concrete definition of the evaluation protocol (train/test splits, stratification, ...) which I would deem equally important to ensure similar results and enforce best practices during evaluation.


**Weaknesses:**

Making decisions about fraud in dynamic settings can lead to feedback loops as labels are only collected for transactions not deemed fraudulent previously, while denied transactions are likely not evaluated further. I think this should be dicsussed in the paper.

I am not sure that I agree with the approach to enforce separability disparity as this essentially introduces new information that is likely not present in real datasets (noise is correlated with group and label). Additionally the effect of this new information is not easy to quantify. Furthermore, this does not change separability in the other attributes. I think a better approach e.g. based on sampling from the data is needed.

I think the quality of the generated data is not sufficiently explored. I would expect the following things to hold:

Ranking of e.g. classifier performances is close between the real and simulated datasets

Datasets are reasonably similar when projected to a 2-D space (e.g.  using T-SNE)

Marginal Distributions of generated data are close to real data.

By investigating those aspects, you could make a more compelling case.

Another (minor) concern that could be remedied by additional experiments is the omission of variables using feature importance.
I think it would be relevant to show how much performance is lost by dropping those variables in order to get a feeling of how strongly this changes the optimization problems (if the dataset is to be used for benchmarking bias mitigation techniques).

**Additional Feedback:**

I think the applicability could be drastically improved by providing access to the trained GAN model allowing the user to sample points. This could e.g. be achieved by providing access to / wrapping  the GAN converted to an ONNX model. This would allow the user to construct additional scenarios for interesting usecases or e.g. allow to simulate systems querying for additional datapoints in a specific subgroup. I can see that this might come with additional privacy concerns.

The 1st and 2nd disparities arer obbtained through ... ?oversampling? why is that required if you have access to a generator. This seems odd to me.

Is there any way to conduct a privacy analysis of the generated data given that you have access to the real dataset, e.g. via k-anonymity? If the source data is differentially private that should mean that any GAN trained on it is also differentially private

The paper should perhaps have a link to the repository. I could not find one in the paper.

**Clarity:**

The paper is generally well-structured and well written.
I think the "FPR Ratio" should be explained more clearly but other then that I could not spot major problems.

**Correctness:**

The dataset is a synthetic version of a private dataset that can not be published due to privacy concerns. In my opinion, the quality of the synthetic data could be more thoroughly investigated (see weaknesses).



**Documentation:**

The paper includes a  datasheet as well as notebooks for replication of the experiments.
I think this could be improved by a more thorough description / providing functionality for good evaluation protocols (data splits, stratification,...) as I would consider this part of a benchmark.

**Ethics:**

There are two main ethical concerns with the presented dataset

Privacy:
The dataset could be used to reconstruct/de-identify private data from individuals, as no sufficient differential privacy guarantees exist.
The authors have made this fact sufficiently clear in the paper, but I do not feel knowledgeable enough in the area to make a recommendation w.r.t. whether this indeed poses a problem.

Misuse:
The dataset could be misused to e.g. develop fraud detection methods. The authors have raised this possibility and warned against this, since models would likely fail in the real world

**Relation To Prior Work:**

I think the section could be improved by e.g. referencing other settings where synthetic data was created for such benchmarks ( if they exist) but other than that, prior work I am aware of is sufficiently discussed.

**Summary And Contributions:**

The paper presents a suite of datasets for bank-account fraud detection based on data generated by a generative model.
The datasets pose an interesting and important challenge for fraud detection as they deviate from other datasets due to strong class imbalance and distribution shifts over time, induced by the dynamic/adversarial nature of fraud detection datasets. Such benchmarks are therefore relevant and pose interesting new challenges for e.g. AutoML systems.

This synthetic nature of the data has the (possible) advantage of better protecting an individual's privacy as only synthetic data is published.
This can however not guarantee the reconstruction of individuals in the data. Two possible downsides are that a) some relevant information might be destroyed in the process of variable selection and GAN fitting and b) the individual's data can not be further enriched in practice because relevant identifiers might be missing.

The paper tackles an important gap, and the approach to generating the data is sound. I would however like to see some additional analysis (see strengths & weaknesses) to be fully convinced that the new datasets indeed provide valuable insights.

---

> ### Author Response · Authors · 2022-08-25
> **Response to reviewer Gpqi (#1)**
>
> Thank you for your comprehensive review, we will address each raised concern in this reply.
>
> > I am however not entirely sure how fairness connects to fraud, I think this should be motivated more thoroughly, e.g. by providing an example scenario where fairness considerations are relevant.
>
> The key idea is that in the context of bank account opening fraud, flagging someone as fraudulent blocks them from accessing banking services, namely credit, which is seen as a basic right by the European Union. A positive prediction of a fraud detection model results in a bank account application being denied. In this context, a false positive is a legit applicant that gets denied access to a banking service. Therefore, this is an application where it is crucial that no particular social group is being discriminated against, i.e., being disproportionately denied access to banking services (for example, one age group having a significantly higher false positive rate).
> Thank you for this comment, we have improved the part where we introduce fraud as being a high-stakes setting (Section 1), and provided a more concrete example (lines 73-74).
>
> > I am missing a concrete definition of the evaluation protocol (train/test splits, stratification, ...) which I would deem equally important to ensure similar results and enforce best practices during evaluation.
>
> We mention that the splits used in the original dataset’s use case were the first 6 months for training and the last 2 months for testing (corresponding to a 80%-20% temporal split). We followed the same split in this paper to stay as close to reality as possible. There was no further stratification, and we have made this clear in the new version of the paper. The performance and fairness metrics used are described at the end of Section 3.1.
> For reproducibility of the fairness-performance study presented in the results section, users should use these splits. However, we also encourage users to experiment with different splits in order to assess how the landscape of performance and fairness changes if the model has access to less training data, for example.
>
> > Making decisions about fraud in dynamic settings can lead to feedback loops as labels are only collected for transactions not deemed fraudulent previously, while denied transactions are likely not evaluated further. I think this should be dicsussed in the paper.
>
> Thank you for raising this concern. The selective labelling problem you describe is indeed commonplace in dynamic settings such as fraud detection. Fortunately, this does not affect our proposed dataset(s), as there was no fraud detection model in production when the original data was gathered.
>
> Further discussion on this topic can be found in this answer to reviewer oXRp:
> https://openreview.net/forum?id=UrAYT2QwOX8&noteId=p3GDisBSPFh
>
>
>
> > I am not sure that I agree with the approach to enforce separability disparity as this essentially introduces new information that is likely not present in real datasets
>
> Thank you for raising this point. The motivation for creating the variants was not so much as to create the most realistic dataset (the most realistic one is the Base dataset), but rather to provide a similar set of data, with an added challenge for unfairness mitigation.
> As we discuss in the paper, another motivation is to provide users with alternative datasets to stress-test, and benchmark unfairness mitigation methods, while keeping the data distribution reasonably realistic.
>
> > - Ranking of e.g. classifier performances is close between the real and simulated datasets
> > - Datasets are reasonably similar when projected to a 2-D space (e.g. using T-SNE)
> > - Marginal Distributions of generated data are close to real data
> > (…)
>
> I think it would be relevant to show how much performance is lost by dropping those variables in order to get a feeling of how strongly this changes the optimization problems
>
> These are great suggestions. We were able to implement your third suggestion and added to the paper. We will include the other suggestions in the camera ready version of the paper.

---

> ### Author Response · Authors · 2022-08-25
> **Response to reviewer Gpqi (#2)**
>
> > I think the "FPR Ratio" should be explained more clearly
>
> We definitely agree and the new version of the paper has a clearer explanation.
> The fairness metric is the ratio between group-wise FPRs, also known as predictive equality. In particular, the denominator is always the highest FPR, and the numerator is the lowest FPR (this means that there is unfairness if any group has a relatively higher FPR and keeps the metric in the range [0,1]).
> The reasoning behind this is that FPR is a measure for the probability of a legit individual to be flagged as fraudulent. For instance, if one group has an FPR of 5%, and another group has an FPR of 10%, legit individuals in the latter group are twice as likely to be wrongly denied access to a bank account, which is unfair. This metric is known as predictive equality, and is considered appropriate for our punitive setting (we have included the appropriate citation for this statement [39] in the new version of the paper).
>
> > I think the section could be improved by e.g. referencing other settings where synthetic data was created for such benchmarks (if they exist)
>
> We have included a part discussing related fraud datasets in section 3.1 (lines 177 to 188), one of which is synthetic. To our knowledge, no synthetic data has been used for benchmarking in the fair ML community.
>
> > I think the applicability could be drastically improved by providing access to the trained GAN model allowing the user to sample points. This could e.g. be achieved by providing access to / wrapping the GAN converted to an ONNX model.
>
> We agree with this point, but unfortunately we cannot share the trained GAN for legal and privacy reasons, as well as the original dataset.
>
> > The 1st and 2nd disparities arer obbtained through ... ?oversampling? why is that required if you have access to a generator. This seems odd to me.
>
> We understand the confusion and we have tried to make it clearer. When we say “oversampling” or “undersampling”, we mean “oversampling/undersampling from the generator”. So we are always referring to sampling from the generator, but we add a prefix to stress that we are increasing, or decreasing the resulting group-wise prevalence distributions, or the size of the groups; in particular, one group may have a relatively lower or higher prevalence in a variant, than in the original dataset; or one group may be larger/smaller in a variant than in the original dataset.
>
> > The paper should perhaps have a link to the repository. I could not find one in the paper.
>
> Thank you, we have added a link to the repository in the new version.

---

### Official Review · Reviewer_oMUf · 2022-07-25
**Tabular dataset synthesized from real fraud datasets to be used for comparing fairness aware machine learning techniques**

**Rating:** 6
**Confidence:** 4
**Clarity:** The paper is easy to follow and clear…

**Strengths:**

The datasets released by the authors are worth getting added to the list of (~15) real world datasets that are commonly used to study fairness-aware machine learning. The datasets are available from GitHub but require the user to download, and then load from the zipped files using a parquet reader.

The authors have taken multiple steps to ensure the protection of real person’s identity in their data generation process. The datasets are generated using CTGAN and any instance that had similarities from the real data were removed to ensure that the datasets don not contain information abut any real person. To help stress test the fairness-aware ML techniques under different biases, the authors provide 5 different varieties of datasets based on varying group, prevalence, and separability biases.

**Weaknesses:**

It is unclear how exactly does the dataset under review adds value to ~15 real world datasets in the research of fairness-aware ML (https://wires.onlinelibrary.wiley.com/doi/full/10.1002/widm.1452). In Section 2.1, the authors merely discuss 3 datasets from related work with missing citations of other related datasets. No empirical comparison of different datasets over the chosen criteria for fairness ML is provided in the submission. Lack of such comparison makes it harder to understand the additional benefit to the fairness-aware ML research community over what already exists today.

A weakness around data accessibility is lack of a standard code to load and transform the datasets from author’s GitHub repository. Both paper and GitHub repository lack a data sample and in order to see a data sample, the user requires to load the datasets using a parquet reader.

Apart from the above, other weaknesses include a lack of contribution in the advancement of data generation techniques or in the advancement of prevention of bias in ML. These two factors are not used for deciding the rating as these factors are not a mandatory requirement for this track. While the novelty in data generation technique is not expected, given the topic of the research, a short discussion and comparison with different datasets and different algorithms used for the prevention of bias can be useful to see the impact of the released dataset.

An additional critique I have for authors (and that is easily modifiable) is that the title and abstract of the paper mention “for fraud detection”, while it is discussed in the paper that the dataset is not relevant for fraud detection studies, but for fairness in ML studies.

The impact of public release of information about the distribution of fraud/non fraud population among different groups is not discussed. For example, the authors mention that no instance in generated dataset can reveal any information about any real users at the individual level. But, at the population level, whether the datasets can reveal the information about fraud that can later be used by attackers to understand patterns of fraud caught by industry, and use that to design new fraud patterns is not discussed.


**Additional Feedback:**

The suggestions are shared in the answers to other review questions. The authors are welcome to share their concern about any feedback, suggestion for improvement or about stated strength/weakness.

**Correctness:**

The datasets are constructed in a sound way. The authors have taken steps to ensure the protection of real person’s identity in their data generation process. The claim about no dataset exists that can meet given selection criteria lacks verification.

**Documentation:**

The documentation about the dataset including the name and definition of variables used is available in data sheet. A sample dataset is lacking in both GitHub repository and in the Appendix. Documentation about the distribution of values of different attributes is also lacking.

**Ethics:**

Even though, an ethical process review was not conducted, the authors had informed the real users about the data collection and usage process. The real user dataset was used to train CTGAN, which was then used to simulate fake instances of real looking users. The fake users do not reveal any individual level information. A discussion about the impact of public release of distribution level information that can be used by fraudsters is lacking.

**Relation To Prior Work:**

The comparison with previous contributions is mentioned in the Section 2.1 but the claims about the added value of given datasets over what is currently used by research community is lacking a detailed justification or/and an empirical analysis. The authors only discuss 3 such datasets from previous contributions.

**Summary And Contributions:**

In the paper, authors share an anonymized and privacy preserving suite of tabular datasets from the field of fraud detection. The authors claim that the datasets can be used for studying fairness of different ML algorithms in practical decision making. Being generated from the field of fraud detection, the datasets fall under tabular data domain, the domain in which, the studies for fairness in ML is lacking, compared to computer vision and NLP tasks.

To support the claim, the authors have discussed about the properties of datasets under 5 different bias conditions and have shown the fairness (in terms of false positive rate ratio of different age based segments) of LightGBM model on each of these datasets. As mentioned in the paper, a good dataset to study the impact of an ML technique on fairness should have certain properties that include the closeness of use case under study with the real world, impact of ML based decisions on people’s life (e.g. criminal justice, financial service etc.), attributes used to model, scale and recency of datasets etc. The commonly used datasets for studying fairness of ML for tabular data do not fit one or more of the given criteria. The datasets generated by authors do satisfy most (if not all) of these criteria and hence are a good fit to study bias in ML for tabular datasets.

The contributions of this submission include public sharing of an anonymized and synthetic dataset from a real fraud detection use case. Even though the datasets in this submission are generated from fraud detection field, they can not be used for any real fraud detection task or to study the effectiveness of ML in fraud detection, due to anonymizations. Rather, the datasets are released to be used for domain independent study and comparison of fairness and performance in dynamic conditions from different ML approaches. The authors have released 5 different variants of the base dataset by adding different types of biases based on sample count or fraud rate in minority and majority groups. The variants are targeted to stress test ML methods for their fairness tradeoffs.

---

> ### Author Response · Authors · 2022-08-26
> **Comment on the advancement of data generation techniques and the prevention of bias in ML**
>
> > other weaknesses include a lack of contribution in the advancement of data generation techniques or in the advancement of prevention of bias in ML.
>
> These are not the main focus of this work. The main focus and contribution of this work (based on the scope of this track) was to release a realistic tabular dataset for fairness evaluation in a fraud detection task. This would be diluted if we included a data generation technique or a bias mitigation method (which by themselves, are complex enough for a separate work). We believe the resources by themselves represent a valuable asset to the ML research community, as the availability and access to realistically large tabular datasets is seldom possible.

---

> ### Author Response · Authors · 2022-08-26
> **Comment on data accessibility**
>
> > A weakness around data accessibility is lack of a standard code to load and transform the datasets from author’s GitHub repository.
>
> Thank you for raising this concern, you are correct. Standardization of the code and better data accessibility are definitive priorities. We have added code for this in the Github repository.

---

> ### Author Response · Authors · 2022-08-26
> **Comment on datasets that meet the selection criteria**
>
> > The claim about no dataset exists that can meet given selection criteria lacks verification.
>
> Thanks for pointing this out. In Section 2.1 of the original paper we discuss how the most popular datasets used for benchmarking fairness do not meet the given selection criteria. We base our critique on our own observations, and on previous literature that analyses these datasets (see references [11-13, 29, 30]).
>
> In the new version of the paper, we have added information on other existing relevant fraud datasets, as well as on other datasets used by the fair ML community (based on reference [31]). We also discuss why these datasets do not meet the selection criteria. For example, none of the fraud datasets contains information on protected attributes, and so cannot be used for fairness benchmarking. Or, among the datasets in fair ML, no single datasets fulfills all the set criteria (e.g., they are too old, not large enough, not based on a specific real-world decision-making task, etc…)
> With this in mind, we believe that the claim is now sufficiently verified.

---

> ### Author Response · Authors · 2022-08-26
> **Comment on how the proposed datasets add value to existing ones in fair ML**
>
> > It is unclear how exactly does the dataset under review adds value to ~15 real world datasets in the research of fairness-aware ML
>
> We thank the reviewer for referencing a pertinent work. It is now included as a reference to this study (as well as the discussed datasets), and is mentioned in lines 49-53. We are aware that there are multiple datasets used for the research of Fairness-aware ML; in the related work we mention the three most common tabular datasets for this practice (according to reference [23]), and talk about their shortcomings.
>
> Additionally, all of the datasets analyzed in the paper referenced by the reviewer (reference [31] in our paper) present one or a combination of factors that limit them considerably in their quality as tools for fairness testing. These factors include, being old, small, having no temporal dimension, not being based on a specific high-stakes decision making task, being easy for models to achieve very high performance, among others.
>
> Conversely, our contribution is a suite of large datasets based on fraud detection data — a high-stakes decision making task where fairness is paramount — posterior to 2015,  that features a temporal dimension, and where each variant presents a unique challenge in terms of performance and fairness. As such, we believe it to be a unique and invaluable contribution to the research community.

---

> ### Author Response · Authors · 2022-08-26
> **Comment on the impact of this release on the distribution of fraud among different groups in the population**
>
> > The impact of public release of information about the distribution of fraud/non fraud population among different groups is not discussed.
>
> With the release of this dataset, it is possible for attackers to study the distributions and replicate successful attempts. This is the main reason why we do not recommend the use of this dataset for real fraud detection applications. In other privately owned datasets, this phenomenon also occurs (albeit in different mediums), as the attackers many times share their successful and unsuccessful strategies online within their groups.

---

> ### Author Response · Authors · 2022-08-26
> **Comment on the paper's title and abstract**
>
> > An additional critique I have for authors (and that is easily modifiable) is that the title and abstract of the paper mention “for fraud detection”, while it is discussed in the paper that the dataset is not relevant for fraud detection studies, but for fairness in ML studies.
>
> Thank you for pointing this out. We have changed the title and abstract to better reflect the role of fraud detection in the datasets (i.e., that they are fraud detection data, but not restricted “for fraud detection studies”). Indeed, the datasets are expected to be used for the measurement of ML performance and fairness in general, as they are based on a real-world, high stakes decision making application.

---

> ### Author Response · Authors · 2022-08-26
> **Thank you for the thorough review**
>
> Thank you for the thorough review. We will address the concerns raised in the following comments.

---

> > ### Comment · Reviewer_oMUf · 2022-08-29
> > **My concerns are addressed**
> >
> > Hi authors,
> >
> > I believe that my concerns are addressed. With that, I update my rating. It would be good to include a sample of the dataset in GitHub repo README.

---

### Official Review · Reviewer_vFM2 · 2022-07-27
**bank-account-fraud**

**Rating:** 5
**Confidence:** 3
**Correctness:** The approach appears reasonable and c…
**Clarity:** The paper is clear and well-motivated.

**Strengths:**

The work addresses a well-motivated problem in a mostly clear manner. Additionally, the authors have developed a nice repository and datasheet.

**Weaknesses:**

I have the following concerns:

- The authors state “_We follow the original strategy for the evaluation of models in the dataset, by 176 training on the first six months of data and validating the models on the last two months._” – What is the reasoning behind this decision / what are potential pitfalls? Could the difference in prevalence between the two time periods be an issue?
- Whether the original relationships between the features and the targets in the generated datasets is preserved or not is unclear. In particular, I was a bit confused by Table 2 in the appendix and what exactly it is showing.


**Additional Feedback:**

Note that in the compressed data link, the download file name is `bank-acount-fraud.zip` as is missing a second “c”.

**Documentation:**

The authors provide a clear repository with notebooks to reproduce their work. Additionally, they provide a datasheet for their datasets.


**Ethics:**

More information in the datasheet on how the original individuals whose data was collected to create the original dataset gave consent may be something to consider (e.g., the Terms and Conditions when signing onto an account).

**Relation To Prior Work:**

The authors clearly outline shortcomings of existing tabular datasets in fairness-related work, such as the Adult, German credit, and COMPAS datasets.

**Summary And Contributions:**

The authors present a collection of tabular datasets. These datasets were generated by applying CTGAN on anonymized bank account application data. Particular subsets contain prespecified bias patterns that the authors enumerate in their work. The authors also provide results for recall and FP ratios in a fairness-blind LightGBM model.

---

> ### Author Response · Authors · 2022-08-25
> **Response to reviewer vFM2**
>
> Thank you for your comprehensive review, we will address each concern raised in this reply.
>
> > What is the reasoning behind this decision / what are potential pitfalls? Could the difference in prevalence between the two time periods be an issue?
>
> This is the train-test split used in the real world use case of the original dataset. However, as we mentioned in the paper, we encourage future users to experiment with different splits as we provide a “month” column to allow for experimentation and studies on fairness-performance tradeoffs in a dynamic setting (which is usually lacking in public tabular datasets).
> Regarding the difference in prevalence: as mentioned in the paper, prevalence does indeed change across time in the original dataset. This is a common phenomenon in fraud detection and other environments, and is something that the employed models have to be robust against.
> As this is a dynamic real-world environment, we are interested in knowing which model would perform the best when deployed in the real-world with accompanying distribution shifts over time. As such, preserving this pattern is crucial to ensure that the dataset is realistic.
>
> > Whether the original relationships between the features and the targets in the generated datasets is preserved or not is unclear. In particular, I was a bit confused by Table 2 in the appendix and what exactly it is showing.
>
> We agree that providing context on the original dataset, and how it compares with the generated one is useful. To this end, we provide some descriptive analysis on how the protected attributes (age and income) compare between datasets in the appendix (section A.5). In particular, how each of these attributes correlate with fraud rate. Furthermore, we show metrics of statistical similarity between the two datasets in section A.3 of the appendix.
>
> As for table 2, we realize that it was not very clear, and have amended it in the new version of the paper, as well as any references made to it in the text.
> It was meant to provide information on the quality of the generative model. The first column shows classifier performance on generated training and generated testing datasets. The second represents performance when training on the generated data, and testing on the original data. The third column represents performance when training on real data, and testing on generated one.
> The idea is that the closer these performance measurements are, the more realistic the generated data is. To illustrate this point, notice how if the data distribution were learned perfectly, then a model trained on it would exhibit the same performance on real testing data, as a model trained on the original training data. This method has been used in the literature (see references [36] and [38]).
> The 4th and 5th columns are metrics of statistical distance between the datasets; the lower they are, the more similar the two distributions, and the higher the quality of the generative model.
>
> > More information in the datasheet on how the original individuals whose data was collected to create the original dataset gave consent may be something to consider (e.g., the Terms and Conditions when signing onto an account).
>
> Thank you for suggesting this. The original dataset is compliant with the European regulation on the collection and storage of user data (GDPR). We have provided all the information we had available to us regarding this type of issue in the datasheet.
>
> > Note that in the compressed data link, the download file name is bank-acount-fraud.zip as is missing a second “c”.
>
> Thank you for spotting this. We have corrected the mistake.

---

> ### Author Response · Authors · 2022-08-25
> **Final comment on reviewer feedback**
>
> We thank the reviewer for pointing out a few aspects of concern.  We have commented and tried to address/improve clarity of the paper on these points. As we’ve mentioned in a final comment to Reviewer “oXRp”, while financial services are of paramount importance to our daily lives, there are almost no publicly available datasets in the domain that gather all the requirements for meaningful performance and algorithmic fairness evaluation. To recap, the proposed suite of datasets — based on a high-stakes decision-making application — is realistic, large, and recent, and, as such, makes for a valuable contribution to the ML research community working with tabular data. With this in mind, we would like to thank the reviewer again for their suggestions, and sincerely encourage them to revise the awarded rating based on the relevance and uniqueness of this resource

---

### Official Review · Reviewer_iuuU · 2022-07-27
**New synthetic dataset for fraudulent bank-opening applications**

**Rating:** 6
**Confidence:** 4

**Strengths:**

This is a real-world bank account opening application dataset, which could complement existing fraud detection datasets. The use of GANs for the development of anonymised datasets with variations is an interesting idea.

[UPDATE: Following the rebuttal, I increased my score from 5 to 6, acknowledging that clarifications and new text (e.g., in related work) have been added. I also welcome the effort of the authors to benchmark more (tree-based) methods, which is in the right direction.]

**Weaknesses:**

Besides clarity, correctness and ethics, which are discussed in separate sections of this review, the presented dataset is biased towards a specific method, which limits its usefulness. In specific:
* Line 201: Different methods would probably return different features as important.
* Line 308: The empirical observations are based on the algorithm used to create the dataset and this may harm generalisation - the same dataset may not work well/similarly for other algorithms.

**Additional Feedback:**

**Questions**
* Does this apply disregarding whether money is stolen from the bank or from another client? Please clarify.
* If the bank covers the expenses of any fraudulent applications (line 154), the societal impact is assumed to be due to false positives (the user is not fraudulent but the bank thinks it is), because everyone should be able to open an account. But it sounds hard to believe that a false positive cannot be solved with minimum effort; either the applicant should be arrested or open the account. Please elaborate.
* 181 How do you motivate your selection of 50 years?

**Comments** (line number shown)
* 57 one "for" should be deleted
* 79 was > were
* Page 7: More informative variant names could help the reader (e.g., I --> disparity).
* Caption of Fig. 1 (4th line): Does "performance" refer to Recall or FPR?
* 344 "the first regards" or "with regards to"
* 551 Placeholder text is not deleted.

**Clarity:**

The paper is well written but some points need to be clarified.
175 The "original strategy" is unclear.
194 This ratio is not clear to me.
202 Random search as a process was not clear to me in this case. On the models? But these should be the best ones. Or were the models trained on random data samples?
230 Is this correlation per feature (real - generated) averaged? If not, please explain.
242 Why did you choose the uniform? Any deviation from the true class ratio could work.
259 It is not clear to me how can you (over/under) sample in a way that only the target group is affected and no other. How is this verified?


**Correctness:**

The claim for feature reduction is not reasonable, the selection of the evaluation methods is not clear and the evaluation design is problematic. In specific:
* Line 200: Removing features which are not important to an algorithm isn't necessarily reducing privacy issues.
* Line 187: Defining a threshold imposed by clients encodes subjectivity, allowing any study to define a different threshold. Also, I find it hard to believe that the clients agree on a single value. At least AUC should have been reported.
* Line 228: Omitting to test on real data sounds the wrong way to me (and couldn't find previous studies that followed it).
* The models could have been assessed on non-anonymised data, to quantify the performance gap.

**Documentation:**

Based on the description in the paper, reproducibility is not possible (couldn't find answers in the datasheet either):
* Line 217 "Manually applied transformations to some fields" make it impossible to repeat the development.
* Line 204 The information on the exact list of the initial features and the resulting 43 is missing. This is also problematic for the evaluation of the correctness of this study (e.g., were the features revealing privacy information removed?).

**Ethics:**

Line 163: The information provided to the applicant before they grant permission to their data is not clear. Should the applicants be aware that the derived data may be used in such studies? Were they?


**Relation To Prior Work:**

Related work was focused outside fraud detection, which sounds wrong. This paper is about fraud, so the analysed related work of a more general domain, is not that helpful and can be misleading. Furthermore:

* Line 111: "[S]everal documentation issues" is not "same and other" that is stated above (line 101). Where are the other issues mentioned above, regarding sampling, etc.?
* Line 122: Does it or does it mean that existing datasets should first be studied in the light of fairness before we start making new ones?


**Summary And Contributions:**

This paper studies the detection of fraudulent bank-opening applications. The authors created a synthetic dataset by selecting the most important features of a machine learning algorithm (LightGBM) and then using GANs to generate new data. Benchmarking variants were created using under/oversampling or by appending new columns. The same algorithm that was used for the feature reduction step was also used to provide an empirical analysis.

---

> ### Author Response · Authors · 2022-08-25
> **Thank you for the thorough review**
>
> Thank you for the thorough review. We are going to address the raised concerns on separate comments.

---

> ### Author Response · Authors · 2022-08-25
> **On the use of LightGBM and its impact on generalization**
>
> > The empirical observations are based on the algorithm used to create the dataset and this may harm generalization - the same dataset may not work well/similarly for other algorithms.
>
> Thank you for raising this issue.
>
> It is a valid criticism to say that the data generation process may lead LGBM models to perform better than other ML algorithms. However, LGBM models were already the best-performing ones on the original dataset, and we aim to maintain the highest predictive power in the generated features as possible. This “highest predictive power” is measured as the best result for any model (on test data); since the best results come from LGBM, testing on other algorithms would arguably not make a difference.
>
>
> Hyperparameter selection for the generative model was indeed based on empirical evaluations of a LGBM model. Moreover, feature selection was based on LGBM feature importance values. Nonetheless, as shown in Table 4 (new addition to the paper), there is high overlap between the most important features for LGBM and those for other high-performance models (Random Forest and Decision Tree). On the other hand, the training of the generative model (CTGAN), the sampling criteria from the generative model, and the added columns for each dataset variant, are all agnostic to the choice of down-stream classifier.
>
> Our choice of the LightGBM model was based on several points:
> It is a production-ready light-weight variant of Gradient Boosting Machines (GBM);
> the GBM algorithm is generally taken to achieve state-of-the-art results on tabular data  [1, 2];
> It is the best-performing model on the original dataset;
>
> We have clarified our reasoning behind choosing the LightGBM algorithm in the latest version of the paper. We agree that using an ensemble of different algorithms, and averaging their results, would perhaps lead to better generalization.
>
> [1] Borisov, Vadim, et al. "Deep neural networks and tabular data: A survey." arXiv preprint arXiv:2110.01889 (2021).
>
> [2] Shwartz-Ziv, Ravid, and Amitai Armon. "Tabular data: Deep learning is not all you need." Information Fusion 81 (2022): 84-90.

---

> ### Author Response · Authors · 2022-08-25
> **On feature importance**
>
> > Different methods would probably return different features as important.
>
> We understand that different models as well as different feature importance methods may have different outcomes. In this dataset, we used LightGBM as it was consistently the best performing algorithm in our setup. We used the five best configurations to obtain the pool of features for the generative model. To address your comment and understand if this was dependent on the model, we repeated this method for other models, and measured the difference in obtained feature sets with Jaccard Index. Different models showed, in general, reasonably high agreement in terms of most important features, which is a positive finding. This study is now available in the appendix.

---

> ### Author Response · Authors · 2022-08-25
> **On the removal of unimportant features and its impact on privacy**
>
> > Removing features which are not important to an algorithm isn't necessarily reducing privacy issues.
>
> Thank you for catching this wording oversight. As the columns are not raw features but aggregations, removing further columns omits information that could possibly make it easier to reconstruct the raw features. Nonetheless, the privacy of the dataset is not due to reducing the number of features, but due to:
> Only using aggregates of raw features (and not the raw features themselves);
> Not releasing the original dataset, and instead training a GAN to generate data with a similar distribution but without corresponding to any real-world data;
> Afterwards, checking that no GAN-generated row is equal to any row in the original data (a large enough model could simply memorize its training data).
>
> Finally, as per the ethics review, we’re currently looking to introduce differential privacy synthesizers into our setup for added privacy guarantees.

---

> ### Author Response · Authors · 2022-08-25
> **On the use of a single threshold (at 5% FPR) for evaluation**
>
> > Line 187: Defining a threshold imposed by clients encodes subjectivity, allowing any study to define a different threshold. Also, I find it hard to believe that the clients agree on a single value. At least AUC should have been reported.
>
> Thank you for raising this concern.
>
> _Clarification on the use of a single threshold for evaluation_:
>
> The real-world outcome of the ML model’s outputs are binary in nature: either accept or deny the customer’s application. In practice, this is done by thresholding the model’s real-valued scores. Fairness metrics, therefore, are concerned with the real-world disparity between these two possible actions, and its disparate impact on different protected sub-groups of society. In our case, a false positive is the only outcome that harms a legitimate customer, hence the fairness metric is the ratio between group-wise FPR. As the fairness metric inadvertently requires binary model predictions, it is arguably unreasonable to measure performance on different model outputs (e.g., using real-valued scores to measure performance and binary predictions to measure fairness).
>
> Moreover, performance metrics too are concerned with predicting the model’s real-world impact, but to the bank instead of to the bank’s clients. This impact is measured as the amount of fraud (in number of attempts or in currency terms) that was caught by the model, as a percentage of the total amount of fraud (this is the Recall metric). At the same time, denying all applications would stop all fraud, but the bank’s goal is to bring in more legitimate clients. As such, banks are solely concerned with performance at low levels of customer attrition; that is, TPR at low levels of FPR; while ROC AUC measures the classifier’s performance at all levels of FPR. As a curiosity, models are put into production with a threshold aiming for a specific metric value, usually from 1% to 5% FPR (or based on alert-rate), and client SLAs (service-level agreement) are set at this specific metric.
>
>
> _Regarding subjectivity of allowing future studies to use a different threshold_
>
> In this work we decided to replicate the real-world process by which automated account opening decisions are made. The 5% FPR choice is based on real-world requirements, but different banks are likely to choose different thresholds (e.g., when aiming for rapid expansion banks may choose a lower FPR value in detriment of higher fraud losses). This choice is meant as a guideline for future studies on this real-world setting, without limiting future works to only use this exact threshold.

---

> ### Author Response · Authors · 2022-08-25
> **Comment on evaluating models on the original data**
>
> > Omitting to test on real data sounds the wrong way to me (and couldn't find previous studies that followed it). The models could have been assessed on non-anonymised data, to quantify the performance gap.
>
> It may be unclear in the text, but models are also trained with the generated data and tested with the original (real-world) data. This is expanded in section A.3. of the appendix. We will make the text clearer. Thank you for pointing this out.

---

> ### Author Response · Authors · 2022-08-25
> **Clarification on the "original" train-test split**
>
>
> > 175 The "original strategy" is unclear.
>
> This is indeed unclear. The term “original strategy” refers to the strategy employed in the original process for training and evaluating a model to be deployed in a production environment. This work seeks to mimic that real-world process by using 6 months for training and 2 months for testing. This is similar to the commonly employed 75-25 split: 6 out of 8 months for training - 75%, and 2 out of 8 months for testing- 25%.

---

> ### Author Response · Authors · 2022-08-25
> **Clarification on the predictive equality fairness metric**
>
> > 194 This ratio is not clear to me.
>
> FPR values are calculated according to the group (e.g., FPR for the younger and FPR for the older groups). The ratio is then calculated, according to the values of FPR of each group. In this ratio, the denominator is always the highest FPR and the numerator the lowest FPR. We extended the text to make it clearer.
> The ratio, or difference, between group FPRs is known in the literature as predictive equality, and is a commonly used fairness metric (see reference [39] in the paper).

---

> ### Author Response · Authors · 2022-08-25
> **Clarification on the use of Random Search**
>
> > 202 Random search as a process was not clear to me in this case. On the models? But these should be the best ones. Or were the models trained on random data samples?
>
> To clarify, we used random search three times, every time for some sort of hyperparameter search over a grid of hyperparameters:
> 1. Randomly sampled 200 different LightGBM hyperparameter configurations; used the top 5 models in terms of TPR@5%FPR in the original data for the feature selection process described in Section 3.2.
> 2. Randomly sampled 70 different hyperparameter configurations for the GAN model. We chose the best GAN model based on the analysis of the evaluation metrics discussed in Section 3.2. Table 2 in the Appendix (A.3) shows the evaluation of the top 5 GAN hyperparameter configurations, sorted by one of the metrics we used.
> 3. Randomly sampled 200 different LightGBM hyperparameter configurations to be trained and evaluated in the generated dataset (and its variants). The fairness and performance of these models is plotted in Figure 1 of Section 4. This Results section is meant as a preliminary showcase of the fairness-accuracy landscape of models in the various variants in the suite of datasets. We encourage the community to experiment with different models and evaluation methodologies.

---

> ### Author Response · Authors · 2022-08-25
> **On related work for fraud detection**
>
> > Related work was focused outside fraud detection, which sounds wrong. This paper is about fraud, so the analysed related work of a more general domain, is not that helpful and can be misleading.
>
> The main contribution of this work is a fraud detection tabular dataset for fairness and ML performance benchmarking. We believe this resource is useful for the wider ML research community working with tabular data and algorithmic fairness, and not only specifically to the research community working in fraud detection tasks. In the related work, we focus on comparing the dataset with commonly used tabular datasets for Fair ML evaluation. Our intention is not to mislead the readers, but to give a succinct landscape of commonly used datasets for fairness benchmarking. However, as you mentioned, it is also relevant to include information on existing datasets for fraud detection. We have added a paragraph about the subject in the related work (lines 106-116).

---

> ### Author Response · Authors · 2022-08-25
> **Comments on dataset documentation**
>
> > Comments regarding documentation
>
> We provide all the possible shareable information for the dataset. Non-disclosed information is kept private due to business / client privacy / legal reasons. While we agree this hinders the reproducibility of the study, it is unfortunately impossible for us to share the original dataset, or its constituting fields.
>
> We want to make sure that the data generation process is as transparent as possible. Nevertheless, the goal of this work is to share the resources themselves (the dataset variants) and not the method we employed to anonymously share this data. Reproducing this data generation process is not an attainable goal due to business and legal constraints that block us from sharing the original data.

---

> ### Author Response · Authors · 2022-08-25
> **Clarifications on several smaller points raised by the reviewer**
>
> Clarifications on several smaller points raised by the reviewer
>
> > 230 Is this correlation per feature (real - generated) averaged? If not, please explain.
>
> That is correct, this correlation value is averaged for all the features in the dataset. Thank you for pointing this out, we will make it more explicit in the paper.
>
> > 242 Why did you choose the uniform? Any deviation from the true class ratio could work.
>
> It is not clear for the authors what the reviewer means. Can you please expand this comment?
>
> > 259 It is not clear to me how can you (over/under) sample in a way that only the target group is affected and no other. How is this verified?
>
> We can control the number of instances to sample with a given set of characteristics, because we have a pool of instances that is considerably larger than the final dataset sample (i.e. the dataset variants).
> As an example, it is possible to create a sample with N instances of applicants with lower age, with a fraud prevalence of Y%. The remainder of the dataset (to reach the total 1M instances) is populated with instances of applicants with higher age, with a prevalence of Z%.
> This might be a case of bad wording from us. Do you believe “stratified sampling” to be the correct term/ definition? If so, we will update the paper.
>
>
> > Line 111: "[S]everal documentation issues" is not "same and other" that is stated above (line 101). Where are the other issues mentioned above, regarding sampling, etc.?
>
> We did not want to transmit that every issue from the Adult dataset is present on the other datasets, but that some of the issues are shared (e.g. age of the data, poor documentation of some variables) and other new ones also appear.
>
> Indeed, the wording choice is not the best. We will rephrase it for better clarity in the paper.
>
>
> > Line 122: Does it or does it mean that existing datasets should first be studied in the light of fairness before we start making new ones?
>
> Yes, this is definitively one viable direction to consider. These should, however, adhere to the rules of thumb presented in the introduction (“What is a good dataset for ML practitioners?” section). We also do believe that there is room to simultaneously use new datasets and evaluate existing datasets regarding fairness.
>
>
> > Does this apply disregarding whether money is stolen from the bank or from another client? Please clarify.
>
> The dataset is regarding applications for bank account opening. Fraudsters try to impersonate other people (victims of identity theft) in order to get access to a new bank account and associated credit (e.g., credit card). The monetary costs of fraud are supported by the banking company.
>
> > If the bank covers the expenses of any fraudulent applications (line 154), the societal impact is assumed to be due to false positives (the user is not fraudulent but the bank thinks it is), because everyone should be able to open an account. But it sounds hard to believe that a false positive cannot be solved with minimum effort; either the applicant should be arrested or open the account. Please elaborate.
>
> It is true that a false positive can be solved with additional queries. However, the effect of a false positive should not be neglected, as it incurs in additional bureaucratic work from the applicant, as well as a longer waiting time for the access to financial services, which might have a negative impact in the life of the individual. Even then, it is not possible to guarantee that there will not be another false positive, even if the application is reviewed by an expert. Note that a False Positive is blocked from having access to a bank account; the escalation to legal authorities is not expected.
>
> > 181 How do you motivate your selection of 50 years?
>
> The selection of 50 years appears as a good compromise between group size and prevalence difference in the dataset. Higher prevalence differences between protected groups make for a more challenging fairness goal.With this, applicants below the age threshold constitute approximately 80% of the sample and have a 1% chance of committing fraud. Applicants above the age threshold constitute 20% of the sample and have a 2% chance of committing fraud. The age feature is available in a continuous form in the dataset, so we invite future users to explore the disparities among different age groups.

---

> ### Author Response · Authors · 2022-08-26
> **Further analysis on the use of LightGBM and its impact on generalization**
>
> > The empirical observations are based on the algorithm used to create the dataset and this may harm generalization - the same dataset may not work well/similarly for other algorithms.
>
> Adding to the previous comment on this issue, we decided to assess the performance of other algorithms in the original and generated datasets, and compare it to LightGBM to see whether our process favoured the latter somehow.
>
> For each of 4 algorithms (LightGBM, XGBoost, Random Forests, and Decision Trees), we randomly sampled 20 configurations of hyperparameters, and trained and evaluated each on the original data, and then on the generated data. It turns out that the top performing configuration for each algorithm was the same across both datasets, except for decision trees. The top 5 models of each algorithm were also similar across datasets, with at least 3 out of 5 configurations being present in both datasets.
>
> When comparing the general performance of all models, the ranking among algorithms was the same. That is, across datasets, LighGBM and XBoost models exhibited the best performance, followed by Random Forests, and finally Decision Trees. Furthermore, the relationship between LightGBM performance and other algorithms stayed the same across datasets. For instance,  **on both datasets**, the best LightGBM model had roughly the same performance as XGboost, 1.05 times the performance of the best random forest, and around 1.42 times the performance of the best decision tree. When averaging these figures over the top 5 configurations of each algorithm, the conclusions remain the same, with XGBoost actually gaining a slight edge on LightGBM in the generated data.
>
> These findings lead us to conclude first that using LightGBM in a stage of the data generation process did not seem to harm the generalization of performance of other algorithms between original and generated data — the best models of each algorithm on the original data are roughly the best ones on the generated data. Second, that LightGBM was not benefited in terms of performance in the generated data in relation to other algorithms — the relationship between the best performing LightGBM models and models of other algorithms stayed the same between datasets.
>
> We hope this comment addressed your concerns, and thank you again for raising this issue.

---

### Official Review · Reviewer_oXRp · 2022-07-28
**Fraud Detection Dataset for Fairness Evaluation**

**Rating:** 5
**Confidence:** 4
**Clarity:** I found the paper clearly written.

**Strengths:**

The authors create a new and interesting tabular dataset. The dataset is synthetically generated based on a real dataset to conserve privacy.

I agree with the authors that new, well-documented tabular datasets are a very useful contribution to the wider research community in general and the algorithmic fairness community in particular. Although this dataset is domain-specific, I think it can be used by many researchers for evaluation purposes and has the potential to be a significant contribution.

The authors do not share the real dataset but do describe the generative process in detail. The code for sampling from the model is available making the dataset more versatile for other users.

The datasheet describing the dataset is comprehensive and easy to read.


**Weaknesses:**

I have one major concern regarding this paper. I did not find, in the paper or the datasheet, a description of how the fraud label is generated. I can imagine two ways:

1) The dataset includes both accepted and rejected applications, and the fraud label is assigned to applications that were rejected as they were suspected of fraud.

2) The dataset only includes accepted applications and the applications that turned out to be fraudulent are labelled.

Both of these cases will lead to implicit bias. The first is only an assumed ground-truth label, based on a decision-making model that may already be biased. The second has a selection bias.

Ideally, the dataset should contain all applications, a flag to say if the application was rejected due to suspected fraud and another variable to note accepted applications that ended up as fraudulent. You would still have the issue of a missing groud-truth for the rejected applications, but getting that label is unrealistic, as is the case in many real-world applications.

I can appreciate why the authors created synthetic biases for this dataset, but they did not present an investigation of the different biases or error rates that exist in the original dataset, both for ground-truth label and the other variables.

More minor concerns:
-- I did not think the authors needed to spend as much space as they did detailing issues with existing benchmarks. I think that space would have been better used to describe existing datasets within the broad domain (in fraud detection/banking etc) and explain how the new dataset is superior to those.
-- The authors give interesting domain information, for example, stating the fraudsters prefer to put a higher age in the application to receive higher credit but don't connect this information to what can be observed in the data itself.

**Additional Feedback:**

I think a section with a descriptive analysis of the original dataset (distribution of values of the different features, correlations between features) would be quite useful for the reader.

Just a suggestion: if used as a benchmark, I think it will be nice to potentially evaluate fairness on a non-binary decision label. For example, instead of accept / reject, it can be: accept with credit / accept without credit / reject. This may represent a more realistic scenario, and will also be very useful as most fairness benchmarks do not go beyond the binary label.

**Correctness:**

The variables seem to be well constructed, as stated above, I have concerns regarding the target label. A more detailed discussion of how the variables in the original dataset were constructed and any limitations or known errors would also be extremely useful.

**Documentation:**

The datasheet was clear and well written. More details regarding how the variables in the original dataset were calculated or constructed would improve it. In addition, it would be great if the authors can mention how and on what legal basis they gained access to the original dataset.

**Ethics:**

The authors cover the privacy aspect thoroughly and I do not think there is any risk or re-identification.

However, I think aspects around implicit bias and fairness are not sufficiently covered, as detailed above.

**Relation To Prior Work:**

I think mentioning similar existing datasets (in addition to popular benchmarks) and highlighting differences would improve the paper further.

**Summary And Contributions:**

I think this dataset may be adopted by the community as a new tabular fairness benchmark, which will be a very useful contribution. However, I have a major concern regarding potential implicit bias in the original dataset that is not discussed by the authors, which unfortunately means I can not recommend accepting the paper at this point.

---

> ### Author Response · Authors · 2022-08-21
> **A clarification on applicant selection bias concerns**
>
> > I have one major concern regarding this paper. I did not find, in the paper or the datasheet, a description of how the fraud label is generated. I can imagine two ways:
> >
> > 1. The dataset includes both accepted and rejected applications, and the fraud label is assigned to applications that were rejected as they were suspected of fraud.
> > 2. The dataset only includes accepted applications and the applications that turned out to be fraudulent are labelled.
> >
> > Both of these cases will lead to implicit bias. The first is only an assumed ground-truth label, based on a decision-making model that may already be biased. The second has a selection bias.
>
> Thank you for bringing awareness to this point. We will answer this point separately as it is the reviewer’s main concern.
>
> You are definitely right in the sense that there is indeed selection bias in the way the labels were gathered, as there is a wide range of regulatory and business constraints that restrict the type of customers some banks may accept. For instance, anti money laundering (AML) regulations dictate that some customers may not be allowed to open a bank account (or credit line). Other applicants may have been pre-rejected due to business constraints (e.g., targeting only customers with 18+ years of age living in country XYZ). Strictly speaking, this does mean that our ground-truth label pool does not correspond to the entire universe of potential customers. It is, however, a sizable and representative pool of customers that would be expected to be able to open a bank account on any bank, as all banks are subject to the same regulatory requirements and similar business constraints. Gathering labels for pre-rejected customers would be either unrealistic or go against bank regulations - e.g., if some customer is signaled as having a very high chance of conducting money laundering, then the bank cannot legally accept their application, and therefore will never know the true label for that customer.
>
> Moreover, no part of the pre-screening process was due to a previously implemented fraud detection ML model, which means that all customers that passed pre-screening were given a chance to prove to be legitimate or fraudulent. This is admittedly rare in real-world scenarios nowadays, as this kind of lax acceptance leads to very high losses for the bank, until some ML model can be put into production. As such, this dataset arguably covers as wide of a customer pool as possible.
>
> In summary, our setting corresponds to your 2nd bullet point, but, as per the points above, we argue it is unrealistic to provide data (ground-truth or not) on *all* applicants. This weakness is inherent to all real-world bank account opening datasets (which are seldom publicly available), and other financial services’ domains, such as consumer lending applications. However, we maintain that the proposed datasets are representative of the types of financial decisions taken by ML models everyday, and, as such, are a valuable contribution to the research community.
>
> We have extended this discussion at the end of Section 1 (lines 92-100). We hope these additions clear up the concerns on label selection bias, and thank the reviewer for raising this important point as a better informed explanation was indeed missing from the original paper.

---

> > ### Comment · Reviewer_oXRp · 2022-08-29
> > **Response**
> >
> > Thank you for this clarification and for the added discussion in the paper.
> >
> > I fully accept that collecting information on all applicants is not realistic. It is more that I think it is critical to give accurate and complete information to the users of the datasets regarding how the data was generated.
> >
> > Please can you be able to add this, clearly and in detail to the datasheet as well?
> >
> > It will also be great to consider what the error in the fraudulent label is? i.e., is it likely that a proportion of the fraudulent accounts opened were not detected?
> >
> > I know there is little time before the final revision is due, so if you feel that there is not enough time, I will be happy with a commitment to do this for the final version.

---

> > > ### Author Response · Authors · 2022-08-29
> > > **Datasheet update and response**
> > >
> > > Thank you for your response.
> > > We have updated the datasheet to reflect this discussion (see A8), adding the information we shared here on how the labels were collected.
> > >
> > > We also addressed the point you raised on the non-fraudulent observations, explaining how the probability of them being mislabeled is exceedingly low. This may also be found in A8.
> > >
> > > We hope this clarifies your concerns, and thank you for contributing to what is now a more informative datasheet.

---

> ### Author Response · Authors · 2022-08-24
> **Added discussion on datasets of related financial domains**
>
> > I did not think the authors needed to spend as much space as they did detailing issues with existing benchmarks. I think that space would have been better used to describe existing datasets within the broad domain (in fraud detection/banking etc) and explain how the new dataset is superior to those.
>
> We understand this concern, and have incorporated further discussion on more datasets (in the fraud detection domain, and in fair ml research) in the revised version of the paper (lines 108-120). At the same time, it is difficult to compare our datasets to others in the same domain because, to the best of our knowledge, this is the _first_ public dataset on online account opening fraud.
>
> Regarding the broader fraud detection domain, we found two additional relevant datasets which were not discussed previously (and have now been added). However, these do not fit the criteria for the evaluation of fairness, as they do not include any information or column on protected attributes. Moreover, these two datasets are on the transaction fraud domain. While false positives in account opening fraud can significantly hinder a person’s life (with no possibility to open a bank account or to access credit), false positives on transaction fraud are arguably less impactful (a transaction that was wrongly blocked can always be unblocked by communicating with the bank).

---

> > ### Comment · Reviewer_oXRp · 2022-08-29
> > **Response**
> >
> > Thank you for adding this, this is helpful.

---

> ### Author Response · Authors · 2022-08-24
> **Added descriptive analysis of original dataset**
>
> > I think a section with a descriptive analysis of the original dataset (distribution of values of the different features, correlations between features) would be quite useful for the reader.
>
> We agree that providing the context of the original dataset, and how it compares with the generated one is useful. To this end, we provide some descriptive analysis on how the protected attributes (age and income) compare between datasets in the appendix (section A.5). In particular, how each of these attributes correlate with fraud rate. This is of particular importance as different prevalences between groups (a.k.a. base rates, different percentages of fraudulent applications) can significantly influence group-wise FPR and TPR. To further elaborate on this point:
>
> $FPR = \frac{p}{1-p} \cdot (\frac{1}{Precision} - 1) \cdot {TPR}$, where $p$ is the prevalence (or base rate).
> That is, if prevalence is increased, FPR will increase, or TPR decrease, or a combination of both (if Precision is fixed).
>
> Furthermore, we show metrics of statistical similarity between the two datasets in section A.3 of the appendix. We additionally added information about the correlation between the target variable and other features in the original and generated datasets, in a new section of the appendix (A.4).

---

> > ### Comment · Reviewer_oXRp · 2022-08-29
> > **Response**
> >
> > Thank you for this. I will review the new sections of the appendix. Please consider highlighting key points from this analysis in the main paper as well.

---

> ### Author Response · Authors · 2022-08-24
> **Comment on non-binary fraud labels**
>
> > if used as a benchmark, I think it will be nice to potentially evaluate fairness on a non-binary decision label. For example, instead of accept / reject, it can be: accept with credit / accept without credit / reject. This may represent a more realistic scenario, and will also be very useful as most fairness benchmarks do not go beyond the binary label.
>
> In the context of this dataset, all applications were accepted with credit. Debit-only account openings generally have a more lax acceptance criteria, as there is limited possibility for fraud. Nonetheless, this is an interesting idea for future work (more specifically in multi-class or multi-label scenarios), thank you for pointing this out.

---

> > ### Comment · Reviewer_oXRp · 2022-08-29
> > **Response**
> >
> > Thank you for this clarification, I think it critical to add this information to the datasheet (that all applications were accepted with credit). This also has some implications when it comes to fairness, as while opening a bank account is a basic right, one can argue that credit is not.

---

> > > ### Author Response · Authors · 2022-08-29
> > > **Follow-up response**
> > >
> > > Thank you for the response. We've added this information to the datasheet (on all applications being accepted with credit) (answer A8).
> > >
> > > Thank you for raising this point on the real-world implications of access to a bank account versus access to credit. To clarify, the fraud detection model blocks both access to a bank account and its line of credit. Nonetheless, unequal access to credit is a known source of discrimination that particularly affects underprivileged groups of society, as credit could also serve as an important driver of social mobility [1, 2].
> > >
> > >
> > > [1] Cohen-Cole, Ethan. "Credit card redlining." Review of Economics and Statistics 93.2 (2011): 700-713.
> > >
> > > [2] Brevoort, Kenneth P. "Credit card redlining revisited." Review of Economics and Statistics 93.2 (2011): 714-724.

---

> ### Author Response · Authors · 2022-08-24
> **Summary**
>
> Financial services are of paramount importance to our daily lives, but there are near to no datasets in the domain that gather all the requirements for meaningful performance and algorithmic fairness evaluation. To recap, the proposed suite of datasets — based on a high-stakes decision-making application — is realistic, large, and recent, and, as such, makes for a valuable contribution to the ML research community working with tabular data.
>
> We thank the reviewer for pointing out the lack of discussion around how the labels of each observation were obtained. We hope to have made it clear that the original dataset contains a highly representative subset of the population that gets screened by ML models, as well as trustworthy labels. The selection bias that occurs at the pre-screening phase (no ML model involved) of account-opening fraud detection pipelines is due to regulatory constraints by which all banks have to abide, and so does not constitute a significant shortcoming of our data in particular. Additionally, we have discussed how every customer that passed the pre-screening phase was given a chance to prove they were legitimate, which adds to the trustworthiness of our data’s labels, especially considering that such a thorough process is not common in the industry. The generated data inherits these beneficial properties.
>
> To address the reviewer’s concerns about a descriptive analysis on the original and generated datasets, we have added to the Appendix further information on how the target variable is related to other features in the original, as well as generated data.
> With this in mind, we would like to thank the reviewer again for their suggestions, and sincerely encourage them to revise the awarded rating based on the relevance and uniqueness of this resource.

---

> > ### Comment · Reviewer_oXRp · 2022-08-29
> > **Response**
> >
> > Thank you for taking the time to consider and react to my feedback. I will be reviewing the new sections of the appendix and will revise my review. Please take note of my requests below for adding these clarifications to the datasheet and confirm if you are happy to address this?

---

> > > ### Author Response · Authors · 2022-08-29
> > > **Response**
> > >
> > > Thank you again.
> > >
> > > We have updated the datasheet, and addressed your concerns in additional comments.

---

### Review · Ethics_Reviewer_gi6o · 2022-08-22

**Recommendation:** 2

**Ethics Documentation:**

Information on consent is missing, see above.

**Ethics Review:**

I read this paper with interest, and agree that this dataset, if published / disseminated through NeurIPS, would fill an important gap in the validation of fairness of ML models.  I'm bringing this up as an ethics reviewer because there is clearly a benefit to having this work be published in NeurIPS.  This being said, there are several risks that were already identified by the the authors and/or reviewers.  Most substantially, the risks that can lead to harms related to privacy violations.

1) Differential privacy (DP) was not used - the model was trained on anonymized data but with no noise added, and so, in principle, a re-identification risk exists.  Designing such a re-identification attack is non-trivial (and it's not the job of the authors to do that).  But the statement they make in their paper and in the datasheet, that no individuals cannot be re-identified, is simply incorrect.

2) The first concern is coupled with the lack of information about how users whose data is included in the original  dataset were consented.  I suspect (because details are not given) that they were not explicitly asked to consent to this type of re-use of their data.  Furthermore, it is unlikely that these users can be reached again, to give further consent to cover this case.

For this reason, I strongly suggest that the authors re-generate the data by training their CTGAN model in a differentially private way, and drawing new synthetic datasets from that DP model.  The authors may want to see https://arxiv.org/abs/2112.09238 and https://arxiv.org/abs/2011.05537 for options on how to do this.

---

> ### Author Response · Authors · 2022-08-26
> **Re-generation of data with differential privacy under way**
>
> Thank you for your review. We are glad you have seen the value of this resource to the ML research community and we are following your recommendations. We believe that these recommendations are feasible to implement in due time and their application will not harm the validity and usefulness of the proposed resources.
>
> The original collected data went through a thorough process of anonymization and feature engineering before being used to train fraud detection models, and later our GAN model.  This, coupled with our additional efforts towards privatizing the data generation process, gives us confidence to assert that the datasets will pose no danger to user privacy.
>
> Nonetheless, we are following your recommendation of data re-generation to ensure that the proposed datasets suite is differentially private. At this moment we are experimenting with a few DP tools that can be applied to the original dataset (smart noise addition) or during the training of the GAN (e.g., DP-GAN). The resources will be updated as soon as possible. Thank you for pointing out useful resources to this end.
>
> User consent for data collection, automated data processing and data transformation was granted in each application (as stated in the datasheet).

---

> ### Author Response · Authors · 2022-08-29
> **Follow-up response**
>
> Again, we thank you for the thorough review. We understand and agree that one of the critical points of this work is the privacy risk of re-identification. This will be the focal point of this comment.
>
> As a direct response to this risk, the originally presented datasets are no longer accessible. Instead, we share noise-injected datasets, following the method presented in [1], using the Laplacian noise mechanism. We explore the utility-privacy trade-off for this method and resulting datasets in appendix.
>
> We also thank the reviewer for the pointed resources; we experimented generating datasets from scratch using DP-CTGANs and PATE-CTGANs with the same hyperparameterization as the best originally used CTGAN. These models, however, failed to converge and produced sub-optimal results. For example, these generative models suffered from mode collapse and were not capable of generating fraudulent examples. This raises the necessity of running hyperparameter optimization studies over the new hyperparameters introduced by these models (more specifically the privacy budget, “epsilon”), and of testing different approaches, such as marginal methods [2]. With these studies, we will strive for datasets with lower budgets of privacy combined with higher ML utility, maintaining the presented metrics in our study. From our experiment (as observed in the paper) these studies are computationally intensive, and might take some weeks to develop and run.
>
> Should, however, any of the datasets obtained from these studies lead to a better privacy-utility trade-off than the datasets obtained with the Laplacian noise mechanism, we will update the published data accordingly, and make sure the camera-ready version of the paper showcases this analysis.
>
> [1] Mivule, Kato. (2012). Utilizing Noise Addition for Data Privacy, an Overview. In Proceedings of International Conference on Information and Knowledge Engineering (IKE 2012), Las Vegas, USA, Pages 65-71.
>
> [2] Tao, Y., McKenna, R., Hay, M., Machanavajjhala, A., & Miklau, G. (2021). Benchmarking differentially private synthetic data generation algorithms. arXiv preprint arXiv:2112.09238.

---

### Meta-Review · Area_Chair_PQRV · 2022-09-12

**Recommendation:** Accept
**Confidence:** 2

**Metareview:**

This contribution releases synthetic data generated by a GAN from a real-world bank fraud dataset to provide 6 datasets (each with 1M instances) for studies of machine-learning fairness. The submission generated many discussions between authors and reviewers. It could play an important role as it would contribute to fairness research were there are only few (~15) real-world datasets.

Several issues were raised, such as the adequate consent of the individuals behind the data, which was asserted to be GDPR-compliant, and the present of possible bias in the detect bias where the authors claimed no false positive because accounts labeled as fraud are given the chance to prove that they are legitimate.

The released data is a GAN-generated synthetic one and there are questions to how well it reflects the statistical properties of the original data. In this respect it is to be regretted that the paper comes without plots showing the marginals.

---

### Decision · Program_Chairs · 2022-09-16

Accept